# SELF-DISGUISE ATTACK: INDUCE THE LLM TO DISGUISE ITSELF FOR AIGT DETECTION EVASION

## ABSTRACT

AI-generated text (AIGT) detection evasion aims to reduce the detection probability of AIGT, helping to identify weaknesses in detectors and enhance their effectiveness and reliability in practical applications. However, existing evasion methods still require high training costs due to fine-tuning and result in text quality reduction owing to the text modification. To address these challenges, we propose Self-Disguise Attack (SDA), a novel approach that enables large language models (LLMs) to actively disguise their output, reducing the detection probability of AIGT. The SDA comprises two main components: the adversarial feature extractor and the retrieval-based context examples optimizer. The former generates disguise features that enable LLMs to understand how to produce more human-like text. The latter retrieves the most relevant examples from an external knowledge base as in-context examples, further enhancing the self-disguise ability of LLMs and mitigating the impact of the disguise process on the diversity of the generated text. The SDA directly employs prompts containing disguise features and optimized context examples to guide the LLM in generating detection-resistant text, thereby reducing resource consumption. Experimental results demonstrate that the SDA effectively reduces the average detection accuracy of various AIGT detectors across texts generated by three different LLMs, while maintaining the quality of AIGT.

## 1 INTRODUCTION

Large language models (LLMs) such as deepseek (DeepSeek-AI, 2024), GPT-4 (OpenAI, 2024) and Qwen (Yang et al., 2024a) have made a profound impact on both industrial and academic fields. Despite their impressive performance, LLMs have also raised significant concerns regarding their potential misuse, such as fake news (Su et al., 2024; Hu et al., 2024), academic dishonesty (Wu et al., 2023; Zeng et al., 2024) and deceptive comments (Mireshghallah et al., 2024). In response to these emerging threats, there is an increasing emphasis on developing robust and reliable methods for detecting AI-generated texts (AIGT) (Song et al., 2025; Bao et al., 2025; Soto et al., 2024).

Existing AIGT detection methods can be broadly categorized as statistical-based methods (Mitchell et al., 2023; Yang et al., 2024b; Verma et al., 2024) and classifier-based methods (Tian et al., 2024; Huang et al., 2024; Guo et al., 2024). These methods have demonstrated remarkable detection performance. As detection capabilities of current AIGT detection methods improve, researchers have also begun exploring evasion techniques to better understand potential weaknesses in AI detectors and enhance their robustness before real-world deployment (Krishna et al., 2023; Zhou et al., 2024). The evasion detection methods aim to decrease the detection probability of AIGT. As illustrated in Figure 1, they can be classified into two categories: the modification-based methods and the generation-based methods. The first category modified the generated text directly to evade detection, typically by leveraging common text attack techniques such as paraphrasing (Krishna et al., 2023) and synonym replacementZhou et al. (2024); Wang et al. (2024). While these methods can effectively bypass AIGT detectors, especially when the target detector is accessible, they often come at the cost of degraded text quality and increased computational overhead, as each text instance requires additional processing. The generation-based methods were designed to enable LLMs to directly generate detection-resistant text through techniques such as fine-tuning (Nicks et al., 2023; Wang et al., 2025) or manually designing prompts (Lu et al., 2023). Although the existing generation-based methods have demonstrated remarkable performance in evading detection,

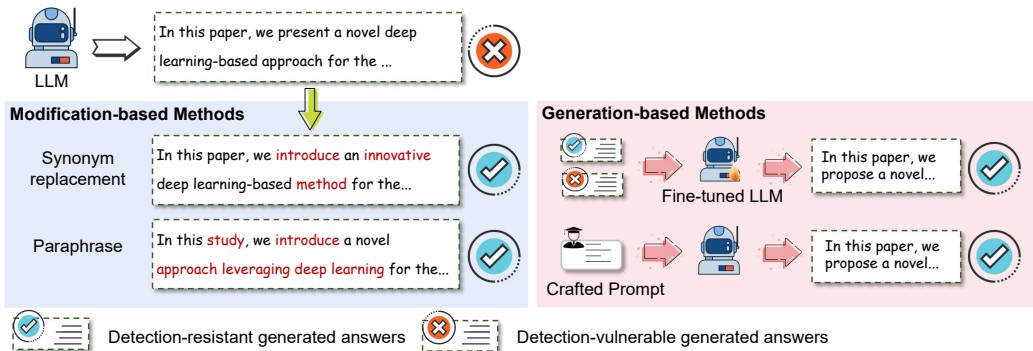

Figure 1: Overview of evasion detection paradigms. (1) The modification-based methods modify the generated text directly. (2) The generation-based approaches guide the LLM to generate detection-resistance text through fine-tuning and manually designing prompts.

several challenges remain: (1) Using fine-tuning techniques results in increased resource consumption. (2) Using prompts to control the output of LLMs limits the diversity of the generated text. Therefore, there is an urgent need to develop more efficient and practical solutions to evade text detection.

To address the above challenges, we propose Self-Disguise Attack (SDA), a novel approach that enables the LLM to actively disguise its output, making it harder to be identified by detectors. SDA integrates two core components: an adversarial feature extractor and a retrieval-based context examples optimizer. The adversarial feature extractor is designed to capture the disguise features that make AIGT more similar to human-written text (HWT). It consists of a feature generator, a text generator, and a proxy detector. Both the feature generator and the text generator are black-box LLMs accessible via API. Under the guidance of capturing disguise features through the adversarial process among the three modules, LLMs are able to understand how to generate more human-like text. These features, described in natural language, also reveal the current linguistic characteristic differences between HWT and AIGT, providing a new perspective for distinguishing between them. We empirically find that these disguise features generated by Qwen-max (Yang et al., 2024a) also help other target LLMs produce more human-like text, highlighting an inherent connection between the outputs of different LLMs. Inspired by retrieval-augmented generation (RAG) (Lewis et al., 2020), we then design a retrieval-based context examples optimizer to enhance the diversity of generated text. It leverages similarity-based retrieval to select the top-$k$ detection-resistant context examples most relevant to the users query from the external knowledge base to guide LLMs in generating the corresponding response. Through this adaptive context example selection strategy, we further enhance the ability of different LLMs to generate detection-resistant text, while preserving the quality of AIGT. We evaluate the performance of the proposed SDA against virous AIGT detectors on three LLMs. Experimental results demonstrate that SDA surpasses four baseline methods, achieving the best evasion detection performance. Our main contributions can be summarized as follows:

- We propose Self-Disguise Attack (SDA) that effectively guides the LLM to disguise its output in order to decrease detection probability. Experimental results demonstrate that our proposed method reduces the average detection accuracy of texts generated by three target LLMs across four detectors, while maintaining the quality of AIGT.

- We design an adversarial feature extractor to capture the disguise features, which can effectively guide the LLM to understand how to generate detection-resistant text. These features also reveal the current linguistic characteristic differences between HWT and AIGT, providing an interpretable new perspective for distinguishing between them.

- Inspired by RAG, we design a retrieval-based context examples optimizer. By constructing an external knowledge base containing detection-resistant texts, the optimizer improves the selection of context examples, further enhancing the self-disguise ability of LLMs and mitigating the impact on the diversity of the generated text.

## 2 RELATED WORK

### 2.1 AIGT DETECTION METHODS

These statistical-based methods detected AIGT by leveraging various statistical differences between AIGT and human-written text (HWT). (Mitchell et al., 2023) proposed a text perturbation method to measure the log probabilities difference between original and perturbed texts. (Su et al., 2023) proposed a zero-shot method that measures the log-probability difference between original text and perturbed text using text perturbation techniques, significantly improving AIGT detection performance. These methods detected AIGT by leveraging various statistical differences between AIGT and human-written text (HWT). The classifier-based methods involve training models on large labeled datasets to detect AIGT. (Tian et al., 2024) introduced a length-sensitive multiscale positive-unlabeled loss, which enhanced the detection performance for short texts while maintaining the detection efficacy for long texts. (Verma et al., 2024) proposed Ghostbuster, a method that processes documents through a series of weaker language models, conducted a structured search over possible combinations of their features, and then trained a classifier on the selected features to predict whether the documents were AI-generated. (Guo et al., 2024) proposed a multi-task auxiliary, multi-level contrastive learning framework, DeTeCtive, which detects generated text by learning different text styles. Existing detectors were capable of achieving high detection performance, which placed higher demands on evasion detection methods.

### 2.2 DETECTION EVASION METHODS

To reveal vulnerabilities in AI detectors before they are deployed in real-world applications, (Krishna et al., 2023) conducted a paraphrasing attack method named discourse paraphraser (DIPPER). This method fine-tuned T5 (Raffel et al., 2020) by aligning, reordering, and calculating control codes, enabling the model to perform diverse modifications and paraphrasing of text without altering its original meaning. (Zhou et al., 2024) proposed humanizing machine-generated content (HMGC). HMGC calculated the importance score for each token, and employed a masked language model to perform synonym replacement on the token with highest score until it can not be detected by the proxy detector. (Wang et al., 2025) proposed Humanized proxy attack (HUMPA), which fine-tunes a proxy small model using reinforcement learning (RL) and modifies the output probabilities of the target LLM based on the probability changes before and after fine-tuning the proxy model. To leverage the capabilities of LLMs, Lu et al. (Lu et al., 2023) proposed the substitution-based in-context example optimization (SICO) method. This method guided the model to generate more human-like text by using samples that have undergone synonym replacement and paraphrasing as in-context examples. Although these methods achieved good performance in evading detection, they were unable to find a trade-off between resource consumption and the quality of the AIGT. Therefore, to reduce resource consumption while safeguarding text quality, we propose the SDA. Its details will be introduced in the following sections.

## 3 METHOD

### 3.1 PROBLEM FORMULATION

Given the user query set $\mathcal{X}$ and an input $x \in \mathcal{X}$, the text generated by LLM $M$ based on $x$ is defined as $t \sim P_M(\cdot \mid x)$. The detection probability of $t$ by detector $D$ is denoted as $p_D(t)$. The goal of the detection evasion task is to construct a function $f$ that systematically adjusts $t$, ensuring that the detection probability $p_D(f(t))$ falls below a specified threshold $\sigma$. The task can be formulated as:

$$\tilde{t} = \arg \min_{t'=F(t)} p_D(t')$$
$$\text{s.t.} \quad p_D(\tilde{t}) < \sigma, \mathcal{D}(t, \tilde{t}) \leq \epsilon, \tag{1}$$

where the function $\mathcal{D}(t, \tilde{t})$ measures the distance between the original text $t$ and transformed text $\tilde{t}$. The parameter $\epsilon$ controls the allowable modification extent to preserve the readability and semantics of texts.

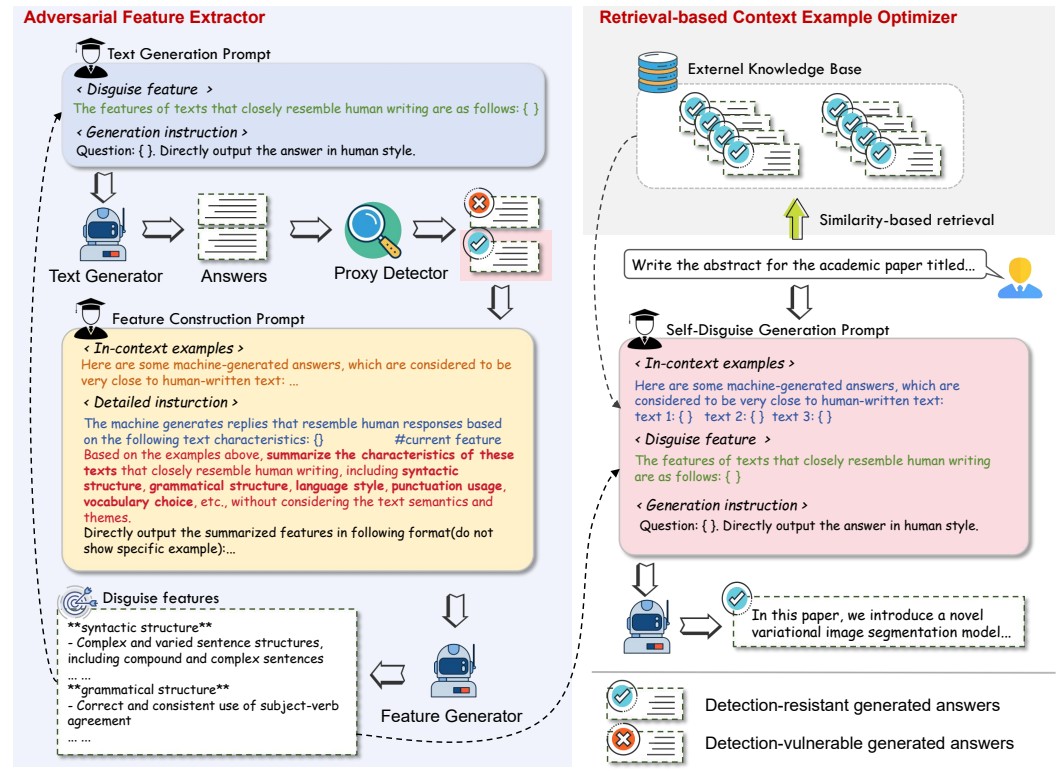

Figure 2: The overall framework of SDA. It consists of an adversarial feature extractor and a retrieval-based context examples optimizer. The former generates disguise features, and the latter identifies the most relevant context examples.8 During generation, both selected context examples and the extracted disguise features are integrated to guide the LLM in generating detection-resistant text.

In this work, our goal is to design a prompt $z$ such that, for any given user input $x \in \mathcal{X}$, the text $t_z \sim P_M(\cdot \mid x \oplus z)$ remains undetectable, where $\oplus$ denotes the concatenation of the user input and the optimized prompt. Formally, our goal is defined as:

$$z^* = \arg\min_z \mathbb{E}_{t_z \sim P(\cdot|x \oplus z)}[p_D(t_z)]$$
$$\text{s.t.} \quad p_D(t_z) < \sigma, \forall x \in \mathcal{X}. \tag{2}$$

## 3.2 SELF-DISGUISE ATTACK

In this work, we propose a novel method, Self-Disguise Attack (SDA), to address the challenges of evading AIGT detection effectively. The overall framework of SDA is shown in Figure 2. The SDA includes two main components: an adversarial feature generator and a retrieval-based context examples optimizer. The feature extractor captures key disguise features that enable LLMs to understand how to generate more human-like text. The retrieval-based optimizer further enhances evasion performance by searching an external knowledge base to identify the most relevant successfully disguised sentences to serve as context examples for the user query. Finally, we design a self-disguise generation prompt to guide the target LLM in generating detection-resistant text during the. As illustrated in Figure 2, the final prompt consists of: (1) context examples selected by the optimizer based on user query, (2) extracted disguise features , and (3) a generation instruction.

## 3.3 ADVERSARIAL FEATURE EXTRACTOR

The adversarial feature extractor comprises a text generator, a feature generator, and a proxy detector.

**The text generator** is a black-box LLM accessed via API, responsible for generating text using the text generation prompt, which includes a generation instruction as mentioned above and the current disguise features as shown below: "*The features of texts resembling human writing are as follows:* { }." The current disguise features are set to empty in the initial stage. In our experiments, we use Qwen (Yang et al., 2024a) as the text generator to generate the candidate text.

**The feature generator** is a black-box LLM accessible via API, responsible for generating disguise features composed of various textual characteristics. To guide the generation of disguise feature, we design a *feature construction prompt*. As shown in Figure 2, it consists of two main components: in-context examples and a detailed instruction. *(1) In-context examples.* We first utilize the proxy detector to evaluate the text generated by the text generator. Those identified as detection-resistant are then collected as contextual examples to construct feature construction prompts, guiding the feature generator in producing disguise features. *(2) Detailed instruction.* Previous studies (Doughman et al., 2025; Soto et al., 2024) have shown that AIGT and HWT exhibit differences in various textual characteristics. To bridge the gap, we explicitly specify the textual characteristics, including syntactic structure, grammatical structure, language style, punctuation usage and vocabulary choice, that the disguise features should focus on in the detailed instruction. Additionally, to eliminate the influence of text semantics, we require the feature generator to disregard the thematic content when generating features.

**The proxy detector.** To refine feature optimization, we employ an existing AIGT detector as a proxy detector to identify whether the generated text is AI-generated.

The complete process of the algorithm is shown in Appendix B. Specifically, we define an optimized step $\eta$ and a maximum error range $\delta$. For each query $x \in \mathcal{X}$, we first use the text generator to generate the corresponding response. The generated texts are then detected using a proxy detector. We collect texts that successfully bypass detection as context examples. Once $\eta$ context examples are collected, they are provided to the feature generator for updating the current features. The text generator then continues generating text based on these refined features. The process above is repeated to extract the critical disguise features. Notably, if the number of AI-detected texts in two consecutive iterations does not exceed $\delta$, the loop ends. Our approach does not require the incorporation of HWT, thereby effectively mitigating the risk of data contamination caused by the widespread use of LLMs during the collection of HWT. Additionally, this feature explains the linguistic differences between AIGT and HWT, providing a new interpretable perspective to distinguish between the two. The complete features we extracted and and the analysis of the rationality of these features are shown in Appendix D.

### 3.4 RETRIEVAL-BASED CONTEXT EXAMPLES OPTIMIZER

Inspired by RAG (Lewis et al., 2020), which enhances the accuracy and richness of LLM by utilizing external knowledge bases, we design the retrieval-based context examples optimizer to further mitigate the limitations of disguise feature prompts on text generation and enhance the self-disguise capability of LLMs. The optimizer comprises two key processes: the construction of an external knowledge base and similarity-based retrieval.

**The construction of external knowledge base.** For each query $x \in \mathcal{X}$, the corresponding undetectable text $y$ is generated by applying the optimal disguise features obtained through the adversarial process, forming a knowledge pair $(x, y)$. The set of knowledge pairs $(\mathcal{X}, \mathcal{Y})$ is stored as the external knowledge base. Then, we leverage a pre-trained model $f$, such as RoBERTa (Liu et al., 2019) to vectorize $\mathcal{X}$, which is difined as:

$$\mathcal{V} = f(\mathcal{X}) \tag{3}$$

We store $\mathcal{V}$ in a vector database, enabling efficient retrieval in subsequent steps.

**Similarity-based retrieval.** For a given user input $x^*$, we first vectorize it into $\boldsymbol{v}^*$:

$$\boldsymbol{v}^* = f(x^*) \tag{4}$$

Then, we query the vector database to retrieve the top-$k$ closest vectors to $v^*$ using the L2 norm:

$$\mathcal{V}_k = \arg \min_{\substack{\mathcal{S} \subset \mathcal{V} \\ |S| = k}} \sum_{\boldsymbol{v}_i \in \mathcal{S}} \|\boldsymbol{v}^* - \boldsymbol{v}_i\|_2 \tag{5}$$

Table 1: Accuracy of different evasion methods across three LLMs. 'M-based' and 'G-based' in the table refer to modification-based methods and generation-based methods, respectively. The optimal results are highlighted in bold in the table.

| LLM | Method | | Detector | | | | Average |
|---|---|---|---|---|---|---|---|
| | | | RADAR | DeTeCtive | MPU | ChatGPT-Detector | |
| Qwen-max | M-based | Paraphrase | 34.50 | 99.00 | 36.50 | 8.00 | 44.50 |
| | | DIPPER | 80.00 | 99.50 | 50.00 | 19.50 | 62.25 |
| | | HMGC | 47.00 | 100.00 | 80.50 | **0.00** | 56.88 |
| | G-based | SICO | 100.00 | 100.00 | 66.00 | 19.00 | 71.25 |
| | | SDA (ours) | **34.00** | **81.00** | **33.00** | 21.50 | **42.39** |
| llama3.3-70B | M-based | Paraphrase | **60.00** | 99.50 | 62.50 | 14.50 | 59.13 |
| | | DIPPER | 89.00 | 96.50 | 61.50 | 30.00 | 69.25 |
| | | HMGC | 60.50 | 100.00 | 82.50 | **0.00** | 60.75 |
| | G-based | SICO | 95.00 | 100.00 | 61.50 | **0.00** | 64.13 |
| | | SDA (ours) | 62.50 | **91.50** | **21.50** | 11.50 | **46.75** |
| deepseek-v3 | M-based | Paraphrase | 51.00 | 99.50 | 36.50 | 5.00 | 48.00 |
| | | DIPPER | 77.00 | 98.50 | 49.00 | 17.50 | 60.50 |
| | | HMGC | 58.00 | 100.00 | 38.00 | **0.00** | 49.00 |
| | G-based | SICO | 92.00 | 99.50 | 26.50 | **0.00** | 54.50 |
| | | SDA (ours) | **22.00** | **75.50** | **6.50** | 3.00 | **26.75** |

where $\mathcal{V}_k$ represents the set of the top-$k$ closest vectors to $\boldsymbol{v}^*$. The knowledge pairs $(\mathcal{X}_k^*, \mathcal{Y}_k^*)$ corresponding to $\mathcal{V}_k$ are retrieved from the external knowledge, and the set of top-$k$ closest undetectable texts $\mathcal{Y}_k^*$, is used as context examples for the target LLM. The complete process of the algorithm is shown in Appendix C.

## 4 EXPERIMENTS

### 4.1 EXPERIMENTAL SETTINGS

**Dataset.** We randomly sample 1000 instances from the human-written texts in the RAID dataset (Dugan et al., 2024). The title field of each sample is then extracted and used to construct the user query set $\mathcal{X}$ in the same manner. For instance, in the case of abstract texts, the corresponding prompt is formulated as: *"Write the abstract for the academic paper titled 'title'."* The dataset is then split into training, validation, and test sets in a 6:2:2 ratio. Notalbly, training set here is used for disguise feature extraction and the external knowledge base construction.

**AIGT detectors.** We employ four AIGT detectors to evaluate the performance of SDA: *(1) ChatGPT-detetcor* (Guo et al., 2023) is a Roberta-based model fine-tuned on the text generated by GPT-3.5. *(2) Radar* (Hu et al., 2023) is a robust detector based on adversarial learning. *(3) DeTeC-tive* (Guo et al., 2024) introduces a multi-task-assisted hierarchical contrastive learning framework to distinguish writing styles of different authors for detecting AIGT. *(4) MPU* (Tian et al., 2024) introduces a multi-scale positive-negative labeling (MPU) training framework, which significantly enhances short text detection performance without compromising long text detection accuracy.

**Baselines.** We employ four evasion methods, including both modification-based and generation-based techniques, as baselines to assess the performance of SDA. *(1) paraphrase* rewrites text by GPT-3.5. *(2) DIPPER* (Krishna et al., 2023) rewrites text by fine-tuning T5. *(3) HMGC* (Zhou et al., 2024) employs adversarial strategies to replace key terms critical for detection with synonyms. *(4) SICO* (Lu et al., 2023) guides the model to generate more human-like text by using samples that have undergone synonym replacement and paraphrasing attacks as in-context examples.

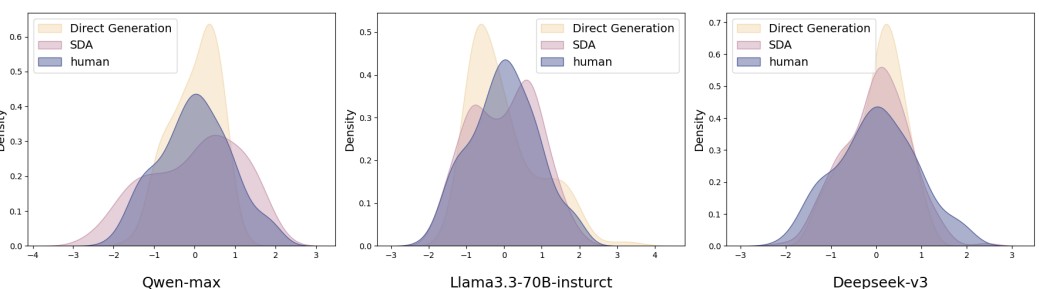

Figure 3: Visualization of text distribution generated by three LLMs before and after Using SDA.

**Evaluation Metrics.** We employ various evaluation metrics to provide a comprehensive assessment of performance of SDA . *(1) Accuracy.* We evaluate the evasion detection effectiveness of the method by testing the detection accuracy of different detectors on 200 attacked texts. *(2) Cosine Similarity.* We assess the performance of model by evaluating the similarity between the generated text and HWT. *(3) Perplexity (PPL).* It measures the level of uncertainty of a given document. A lower value is usually associated with higher quality in the generated text. *(4) Self-BLEU.* It is a metric used to measure the diversity of generated text. A lower value indicates higher diversity in the generated text.

**LLM Selection.** In our experiment, we utilize three latest LLMs, including Qwen-max (Yang et al., 2024a), deepseek-v3 (DeepSeek-AI, 2024), and LLaMa3.3-70B-instruct (Team & Meta-AI, 2024), to validate the effectiveness of SDA. Both the feature generator and the text generator in the adversarial feature extractor utilize Qwen-max. Notably, Different target LLMs are guided by the same disguise features generated by Qwen-max.

**Implementation Setting.** To simulate the real-world detection scenario, we directly use the publicly available version of the corresponding detector on HuggingFace[1] for our experiments. For all experiments in our study, the proxy detector used is ChatGPT-detector, a RoBERTa-based detector that has been widely applied in previous works (Lu et al., 2023; Dugan et al., 2024). Additionally, we set the detection threshold $\sigma = 0.5$, the optimized step $\eta = 5$, the maximum error range $\delta = 2$ and the number of context exampls $k = 5$. All experiments conducted using two NVIDIA RTX 4090 GPUs, each equipped with 24GB of GPU memory.

### 4.2 MAIN RESULTS

We compare our method with four baselines across three different target LLMs. As shown in Table 1, SDA consistently outperforms the other baselines by causing a greater reduction in the average detection accuracy of the four detectors for three target LLMs. For the case of alignment between the object detector and the agent detector (both being ChatGPT-detector), both HMGC and SICO are able to reduce detection accuracy to zero. HMGC is a modification-based attack that iteratively replaces important words in the text until it successfully evades the proxy detector. This ensures that the modified text cannot be detected by the proxy detector, but its performance is not reliable against unknown detectors. SICO, on the other hand, uses manually designed in-context examples to guide the LLM in generating text that avoids detection. We argue that this limits the LLM's ability to learn effective evasion features, resulting in poor generalization when facing unseen detectors. In contrast, our SDA method learns more general disguise features. It aims to reduce the average detection accuracy across all unseen detectors, even when they differ from the proxy detector. As shown in Table 1, SDA demonstrates strong generalization ability. Additionally, we visualize the text generated using SDA across three LLMs, as shown in Figure 3. Compared to direct generation, the texts generated with SDA are closer to HWT, demonstrating the effectiveness of the proposed SDA. We provide a case study in Appendix E.

---

[1]https://huggingface.co/

Table 2: Comparison of the perplexity of texts between SDA and modification-based evasion detection methods.

| LLMs | Qwen-max | llama3.3 | deepseek-v3 |
|---|---|---|---|
| Direct Generation | 21.40 | 15.33 | 23.63 |
| Paraphrase | 29.80 | 23.64 | 30.41 |
| DIPPER | 24.62 | 21.37 | 25.39 |
| HMGC | 22.47 | 16.24 | 24.01 |
| SDA (ours) | **16.68** | **15.25** | **20.47** |

Table 3: Comparison of the text quality between SDA and generation-based evasion detection method.

| LLMs | metric | SICO | SDA(ours) |
|---|---|---|---|
| | PPL ↓ | **16.55** | 16.68 |
| Qwen-max | cosine similarity ↑ | 0.9877 | **0.9946** |
| | self-BLEU ↓ | 0.5510 | **0.4920** |
| | PPL ↓ | 15.30 | **15.25** |
| llama3.3 | cosine similarity ↑ | 0.9816 | **0.9944** |
| | self-BLEU ↓ | 0.5454 | **0.4952** |
| | PPL ↓ | **19.64** | 20.47 |
| deepseek | cosine similarity ↑ | 0.9913 | **0.9977** |
| | self-BLEU ↓ | 0.5084 | **0.4383** |

### 4.3 EVALUATION OF TEXT QUALITY

**Comparison with Modification-based Methods.** Since diversity metrics primarily evaluate the generation capabilities of language models, we focus solely on the perplexity of the attacked text for the modification-based methods. As shown in Table 2, modification-based approaches tend to degrade text generation quality, whereas our proposed SDA effectively enhances the quality of AIGT. Specifically, on Qwen-Max, our approach achieves a perplexity reduction of 4.72 relative to direct generation.

**Comparison with Generation-based Methods.** We evaluate the cosine similarity between the generated text and HWT under different methods, while using self-BLEU to measure the diversity of AIGT. The results are shown in Table 3. However, compared to SICO, the cosine similarity of SDA is higher across all three LLMs, indicating that SDA can effectively mimic HWT. Furthermore, the diversity of text generated using SDA surpasses that of SICO across all three LLMs, suggesting that SDA effectively mitigates the influence of prompts on AIGT diversity. We further conduct human evaluation for these methods across three dimensions: fluency, semantic clarity, and the probability of AI generation, as detailed in Appendix F.

### 4.4 EFFICIENCY EXPERIMENTS

SDA does not require fine-tuning the model, thereby reducing computational resources during training. However, considering the impact of longer prompts on generation time, we compare the time required to generate 50 samples using Qwen-max. As shown in Figure 4, SDA introduces only minimal overhead during inference compared to SICO, demonstrating its efficiency.

### 4.5 ABLATION STUDY

We validate the effectiveness of the two main components, the adversarial feature extractor and the retrieval-based context examples optimizer, through ablation experiments. The test results for different combinations are shown in the table 4. Under the guidance of the proposed SDA, the average detection accuracy of texts generated by the three latest LLMs decreased by more than 14%. Additionally, the experi-

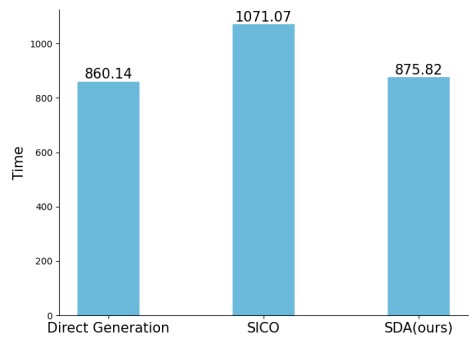

Figure 4: Comparison of generation time.

mental results show that, incorporating the features extracted by the feature extractor significantly reduces the average detection accuracy. Specifically, compared to direct generation, the average de-

Table 4: Ablation study. "Disguise feature" indicates whether the features extracted by the adversarial feature extractor are used. "Retrieval" denotes whether the context examples optimizer is employed. The "✓" indicates that the component is enabled, while "✗" that the component is not employed. The optimal results are highlighted in bold in the table.

| Disguise feature | Retrieval | method | Qwen-max | Llama3.3-70B-instruct | deepseek-v3 |
|---|---|---|---|---|---|
| ✓ | ✓ | RADAR | 34.00 | 62.50 | 22.00 |
| | | DeTeCtive | 81.00 | 91.50 | 75.50 |
| | | MPU | 33.00 | 21.50 | 6.50 |
| | | chatgpt detector | 21.50 | 11.50 | 3.00 |
| | | Average | **42.38** | **46.75** | **26.75** |
| ✗ | ✓ | RADAR | 40.50 | 46.50 | 32.00 |
| | | DeTeCtive | 83.50 | 88.00 | 71.00 |
| | | MPU | 38.50 | 50.50 | 6.00 |
| | | chatgpt detector | 28.00 | 29.00 | 2.00 |
| | | Average | 47.63 | 53.50 | 27.75 |
| ✓ | ✗ | RADAR | 49.50 | 99.50 | 45.50 |
| | | DeTeCtive | 61.50 | 90.50 | 66.50 |
| | | MPU | 48.00 | 14.50 | 0.00 |
| | | chatgpt detector | 21.00 | 1.00 | 0.00 |
| | | Average | 45.00 | 51.38 | 28.00 |
| ✗ | ✗ | RADAR | 36.50 | 48.00 | 56.50 |
| | | DeTeCtive | 98.50 | 100.00 | 90.00 |
| | | MPU | 80.50 | 84.50 | 39.00 |
| | | chatgpt detector | 27.00 | 38.50 | 10.50 |
| | | Average | 60.63 | 61.50 | 49.00 |

tection accuracy decreases by over 10% across all three LLMs when disguise features are included. Furthermore, the decrease in consistency across the three LLMs suggests that the disguise features generated by one LLM can generalize to others, indicating an inherent correlation between the texts produced by different LLMs. Moreover, incorporating a retrieval-based context examples optimizer to refine the selection of context examples can also significantly reduce the average detection accuracy of model. We also conduct additional ablation experiments, including introducing easily detectable samples during the feature extraction and selecting different proxy detectors, to validate the rationality of our approach. The experimental results are shown in Appendix G.1. Notably, we propose a potential defense solution against SDA, as detailed in Appendix H.

## 5 CONCLUSION

In conclusion, we propose a novel detection evasion method SDA that effectively guides the LLM to disguise its output in order to generate more human-like text. The SDA primarily consists of two components: the adversarial feature extractor and retrieval-based context examples optimizer. The former is designed to capture the key disguise features that enable the LLM to understand how to generate human-like text. The latter improves the selection of context examples, further enhancing the self-disguise ability of LLMs and mitigating the impact of the disguise process on the diversity of the generated text. Finally, we construct a self-disguise generation prompt combined with optimized contextual examples and disguise features to guide the target LLM in generating detection-resistant texts. Our extensive experiments on evasion demonstrate the superior performance of SDA, which significantly reduces the detection probability of AIGT while effectively preserving text quality.

ETHICS STATEMENT

The primary objective of this paper is not to provide techniques for evading AIGT detection systems, but rather to expose vulnerabilities in current detection mechanisms. We firmly believe that with increased attention and efforts toward this issue, the research community can devise more sophisticated and effective techniques to enhance the robustness and reliability of machine-generated text detection systems in the face of evolving adversarial threats.

REPRODUCIBILITY STATEMENT

The code and data required to reproduce all experiments has been submitted as supplementary material.

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

## A    LLM USAGE STATEMENT

During the preparation of this manuscript, ChatGPT (OpenAI, 2024) was employed for language polishing and stylistic refinement.

## B    ADVERSARIAL FEATURE EXTRACTION

The algorithm continuously generates and optimizes disguised features to counter the detector, aiming to reduce the detection probability of the generated text. As shown in Algorithm 1, the algorithm begins by initializing an empty list $batch\_success\_es$ and an empty set of $current\_features$. For each data point in the training set, it generates a disguised text using the text generator $M_{text}$ and evaluates it with the detector. If the detection probability of $text$ if smaller than the threshold $\sigma$, it is added to $batch\_success\_es$. Once the number of successful disguises in $batch\_success\_es$ reaches a threshold $\eta$, the algorithm calculates the step range since the last optimization. If the difference between the current step and the optimization step is smaller than the allowed error range $\delta$, it increments the stop counter $stop$. Otherwise, it updates $current\_features$ using the feature generator $M_{feature}$ and resets $batch\_success\_es$ to an empty list. If the stop counter exceeds 2, the optimization process stops, and the final disguised features, $Disguise\ features$, are returned.

## C    RETRIEVAL-BASED CONTEXT EXAMPLES OPTIMIZER

This algorithm, as shown in Algorithm 2, aims to optimize the selection of context examples based on a user's query and an external knowledge base. Specifically, the process first vectorizing the user's query $x^*$ using the pre-trained model $f$. The vectorized query $v^*$ is then used to retrieve a set of relevant context examples from the external knowledge base $B$ based on the similarity between the query vector and the knowledge base. After retrieving the most relevant context examples $\mathcal{V}_k$, the corresponding context examples $\mathcal{X}_k^*$ and their associated labels $\mathcal{Y}_k^*$ are obtained from the knowledge base. Finally, the algorithm returns the selected context examples $\mathcal{Y}_k^*$ as the output.

## D    DISGUISE FEATURES

The final extracted features are shown in Figure D. From the perspective of syntactic structure, recent research (Reinhart et al., 2025; Russell et al., 2025) shows that LLMs tend to overuse certain syntactic and lexical patterns, lacking the stylistic diversity and variability typical of human writing. For example, instruction-tuned models (e.g., Llama-3-Instruct (Team & Meta-AI, 2024)) use present participial clauses at 2–5 times the rate of human writers. Therefore, our disguise features encourage LLMs to adopt more varied syntactic constructions, which has practical significance for reducing detectability. From a linguistic style perspective, (Reinhart et al., 2025) also point out that LLMs, such as GPT-4o, significantly reduce the use of passive voice, with its usage being only half that of human-authored texts. Therefore, our disguise features encourage LLMs to correctly use both active and passive voices, aligning with the observed differences between the two types of texts. Regarding the use of punctuation, (Zamaraeva et al., 2025) pointed out that LLMs tend to use commas and semicolons to separate sentence components, which are less common in human

---

**Algorithm 1** Adversarial Feature Extraction.

---

**Require:** Training data loader $\mathcal{X}_{train}$, Text generator $M_{text}$, Feature generator $M_{feature}$, Detector $D$, Optimized steps $\eta$, Maximum Error range $\delta$, Detection threshold $\sigma$
**Ensure:** $Disguise\ features$
1: $batch\_success\_es \leftarrow$ empty list
2: $current\ features \leftarrow empty$
3: $init\_idx \leftarrow 0$
4: $stop \leftarrow 0$
5: **for** $index,\ x$ in $\mathcal{X}_{train}$ **do**
6:     $text \leftarrow$ Generated by $M_{text}$
7:     **if** $P_D text < \sigma$ **then**
8:       $batch\_success\_es$.append($text$)
9:     **end if**
10:    **if** len($batch\_success\_es$) $\geq \eta$ **then**
11:      $step\_range \leftarrow index - init\_idx$
12:      $init\_idx \leftarrow index$
13:      **if** $step\_range - optimize\_step \leq \delta$ **then**
14:        $stop \leftarrow stop + 1$
15:      **else**
16:        $stop \leftarrow 0$
17:        $current\ features \leftarrow$ Generated by $M_{feature}$ based on $current\ features$ and $batch\_success\_es$
18:      **end if**
19:      $batch\_success\_es \leftarrow$ empty list
20:    **end if**
21:    **if** $stop > 2$ **then**
22:      **break**
23:    **end if**
24: **end for**
25: **return** $Disguise\ features$

---

---

**Algorithm 2** Retrieval-based Context Examples optimizer

---

**Require:** Users query $x^*$, External knowledge base $B$, pre-trained model $f$, The number of retrieved context examples $k$.
**Ensure:** Context examples $\mathcal{Y}_k^*$
1: $v* \leftarrow$ Vectorize $x^*$ using $f$
2: $\mathcal{V}_k \leftarrow$ Retrieve from $B$ based on $v^*$.
3: $(\mathcal{X}_k^*, \mathcal{Y}_k^*) \leftarrow$ Obtained from $B$ based on $\mathcal{V}_k$
4: **return** Context examples $\mathcal{Y}_k^*$

---

writing. Additionally, the use of dashes is more frequent in AIGT than in human text. This aligns with the guidelines for punctuation usage in the disguise feature of this paper.

Overall, the disguise features we extract are consistent with the known linguistic differences between AIGT and HWT, providing interpretability and offering a new perspective for distinguishing the two types of texts.

## E   CASE STUDY

In this section, we provide a case study where GPT-5 is employed to evaluate each sentence with scores ranging from 1 to 10. As illustrated in Figure E, the proposed SDA method attains the best text quality scores.

## F   HUMAN EVALUATION

**Disguise features**

**syntactic structure**
- Complex and varied sentence structures, including compound and complex sentences
- Use of parallelism to enhance clarity and readability
- Appropriate use of conjunctions and transitions to connect ideas

**grammatical structure**
- Correct and consistent use of subject-verb agreement
- Proper use of tenses, with a mix of present, past, and future as appropriate
- Effective use of active and passive voice for clarity and emphasis

**language style**
- Formal and academic tone, suitable for scholarly articles
- Clear and concise, avoiding unnecessary jargon while using technical terms appropriately
- Use of descriptive and precise language to convey complex ideas

**punctuation usage**
- Proper use of commas, periods, semicolons, and colons for clarity and separation of clauses
- Consistent use of quotation marks for titles and specific terms
- Appropriate use of parentheses and dashes for additional information or clarification

**vocabulary choice**
- Use of domain-specific terminology relevant to the field
- Incorporation of sophisticated vocabulary to convey nuanced ideas
- Avoidance of colloquialisms and informal language

Figure 5: The final extracted disguise feature.

we collect 100 texts modified by each attack method and invited three experienced LLM users to rate them on three dimensions: fluency, semantic clarity, and the probability of AI generation. Scores ranged from 1 to 5, where a higher fluency score indicates smoother text, a higher semantic score reflects better interpretability, and a higher AI-probability score suggests a greater likelihood of being identified as AI-generated. As shown in Table 5, the

Table 5: Human evaluation for different methods.

| Method | fluency | semantic | AI-prob |
|--------|---------|----------|---------|
| HMGC | 1.24 | 1.25 | 4.83 |
| DIPPER | 3.77 | 3.86 | 2.63 |
| SICO | 4.06 | **4.09** | **2.21** |
| SDA(ours) | **4.07** | 4.03 | 2.59 |

generative-based method outperforms the modification-based approach in terms of semantics, fluency, and AI-probability. Additionally, Our SDA and SICO demonstrate comparable performance in human evaluations.

## G ABLATION STUDY

### G.1 INTRODUCING EASILY-DETECTED SAMPLES

we introduce easily-detected samples each time we optimize the features and conduct experiments on Qwen-max. As shown in Table 6, the text generated based on this feature had an average accuracy 3.49% higher than the original SDA across four detectors, indicating relatively weaker evasion performance. We believe that mixing these two types of examples can blur the optimization direction, as the model is pulled in conflicting ways. This confusion may lead to a collapse in the feature space, making it harder for the model to capture subtle and informative patterns. Additionally, presenting both types of examples increases cognitive load, which can divert the model's attention away from the hard examples that matter most.

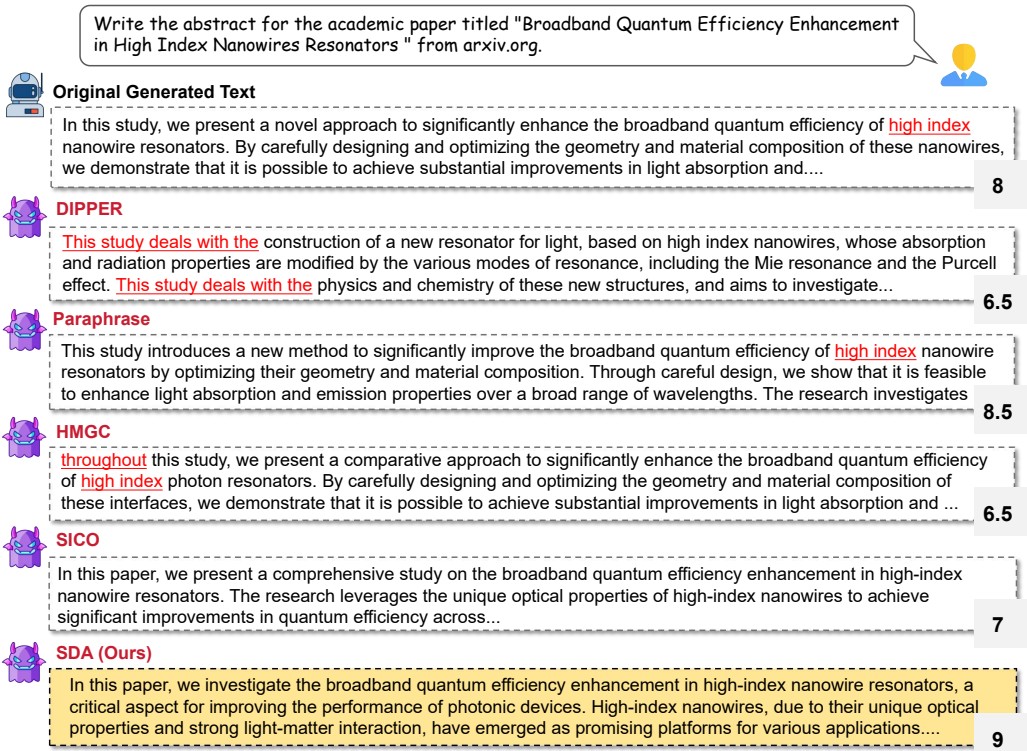

Figure 6: The final extracted disguise feature.

## G.2 DIFFERENT PROXY DETECTOR

we experiment with a more advanced detector, MPU, as the surrogate detector. Results as shown in Table 7 demonstrated that using MPU achieved an average detection accuracy of 44.13% across four target detectors, which is 1.74% higher than when using the ChatGPTDetector. We believe strong detectors shrink the space for optimizing adversarial features because of their strict decision boundaries. This often causes overfitting, so the model ends up learning only narrow, locally optimal features. In contrast, weak detectors, with looser boundaries, allow for more generalizable features to be extracted. Moreover, during training, strong detectors are more likely to hit the $\sigma$ threshold prematurely, resulting in early termination of the optimization process. Weak detectors, on the other hand, provide a smoother and more gradual optimization path, enabling deeper feature exploration.

Table 6: Evaluation results of introducing easily detectable samples.

| Method | $\text{SDA}_{+easy}$ | SDA |
|---|---|---|
| RADAR | 44.00 | **34.00** |
| DeTeCtive | 82.00 | **81.00** |
| MPU | 33.00 | 33.00 |
| chatgpt detector | 24.50 | **21.50** |
| Average | 45.88 | **42.39** |

Table 7: The impact of proxy detector.

| Method | $\text{SDA}_{MPU}$ | $\text{SDA}_{chatgpt-detector}$ |
|---|---|---|
| RADAR | 54.00 | **34.00** |
| DeTeCtive | **78.00** | 81.00 |
| MPU | **26.00** | 33.00 |
| chatgpt detector | **18.50** | 21.50 |
| Average | 44.13 | **42.39** |

## H    POTENTIAL DEFENSIVE METHOD

A simple and intuitive defense method is to collect SDA-based data to train the detector, thereby enhancing its robustness against such attacks. In the future, we will also explore more robust detection approaches.

