# OpenReview forum: "Self-Disguise Attack: Induce the LLM to disguise itself for AIGT detection evasion"
_ICLR.cc/2026/Conference — ICLR 2026 Conference Withdrawn Submission_

### Official Review · Reviewer_VmS7 · 2025-10-15

**Soundness:** 2
**Presentation:** 1
**Contribution:** 2
**Rating:** 2
**Confidence:** 4

**Summary:**

This paper proposes a finetuning-free, generation-based evasion method driven by prompt engineering + detector feedback, termed Self-Disguise Attack (SDA). SDA consists of two parts: (i) an adversarial “disguise-feature” extractor that employs black-box LLMs as a text generator and a feature generator, with a proxy detector (ChatGPT-detector, RoBERTa family) providing feedback to iteratively distill natural-language descriptions of “human-style disguise features”; and (ii) a retrieval-based in-context example optimizer that uses vector search over a self-built external repository to select examples most relevant to the current query and that have previously passed detection. These examples, together with the disguise features, are concatenated into the final prompt to guide the target LLM to produce more “human-like” text and reduce its probability of being detected.

Experiments attack four detectors (RADAR, DeTeCtive, MPU, ChatGPT-detector) across three LLMs (Qwen-max, LLaMA-3.3-70B-Instruct, DeepSeek-V3). Results show that SDA achieves lower average detection accuracy (lower is better) than multiple modification- and generation-based baselines (Paraphrase, DIPPER, HMGC, SICO), while maintaining or improving quality and diversity metrics such as PPL, Self-BLEU, and cosine similarity to HWT. The paper also includes human evaluation and ablation studies. The method requires no finetuning, relies solely on prompting and retrieval, and claims low resource overhead and transferability.

**Strengths:**

1. A training-free evasion approach for AI-generated text detection that relies solely on prompt engineering to produce highly human-like, detector-evading outputs.

2. Better preservation of text quality relative to other baselines.

3. A relatively novel application of RAG.

**Weaknesses:**

**This paper is not well written.** There are multiple citation errors and invalid figure references (see weaknesses below). It will require substantial revisions before publication. In its current form, it is not ready for a top-tier venue like ICLR. The main weaknesses, ranked by severity, are as follows:

1. **Exaggerated performance.** In this paper, you use the ChatGPT-detector as your proxy model, which is based on RoBERTa-base; all the detectors you attack in evaluation are also RoBERTa-base-based. Although you add MPU (also RoBERTa-base-based) as a proxy model in the appendix, I am concerned that using the same backbone family during the training phase (proxy detector) and the testing phase (detectors under attack) leads to overfitting. This is especially problematic when your method’s margins over the baselines are small, which can easily inflate the perceived performance.

2. **Limited evaluation.** The paper evaluates only 4 AI-generated text detectors, and all of them belong to the model-based category. Can your method effectively attack metric-based detectors (e.g., Binoculars, FastDetectGPT, DNAGPT) and watermark-based detectors? In addition, you evaluate only **200** samples from RAID and conduct human evaluation on just **100** texts. I consider this evaluation scope to be limited.

3. **Limited metric reporting.** In Table 2, you report perplexity relative to modification-based evasion detection methods, while Table 3 reports perplexity, cosine similarity, and BLEU versus SICO. Why not report these metrics in Table 2 as well? Moreover, I believe reporting the change in perplexity (before vs. after the attack) would better highlight the quality shift caused by the attack. Table 4 reports generation time, but given that your method heavily relies on in-context learning, you should also report API call costs to support the claim that your approach is more efficient than the baselines.

4. **Numerous citation and figure-reference errors.**

   * **Line 48**: “Zhou et al. (2024); Wang et al. (2024)” should be enclosed in parentheses.
   * **Lines 113, 118, 120, 123, 133, 136, 139**: the parentheses should be removed from these citations.
   * In the Case Study section at **line 751**, you refer to Figure E, but the link points to Appendix E; this figure should actually point to Figure 6.

Given the concerns above, I recommend a rejection of this paper in its current form.

**Questions:**

1. If your proxy model changes, how would your results be affected?
2. How effective is your method against the detectors I mentioned under Weaknesses? I checked the code and found interfaces for detectors such as FastDetectGPT; it appears your method should be applicable to these detectors.
3. Please include a larger number of samples in the evaluation, and add a comparison of API costs.

**Details Of Ethics Concerns:**

This paper proposes an attack method for evading AI-generated text detectors, but it does not provide a detailed discussion of how to detect such evasion—only a brief treatment in the appendix. I recommend that the authors include a more thorough discussion of defense methods to minimize potential negative impacts. In addition, please disclose the detailed protocol for the human review process.

---

### Official Review · Reviewer_yWpt · 2025-10-28

**Soundness:** 2
**Presentation:** 2
**Contribution:** 2
**Rating:** 2
**Confidence:** 5

**Summary:**

This paper proposes a novel Self-Disguise Attack (SDA), which enables large language models (LLMs) to automatically disguise their outputs through carefully designed prompts, thereby evading AI-generated text (AIGT) detectors. The attack consists of a adversarial feature extractor aiming for generating disguise features and a retrieval-based context examples optimizer that selects the most relevant successfully disguised samples from an external knowledge base. Experimental results demonstrate that SDA significantly reduces the success rate of various detectors while maintaining or even improving the overall text quality and diversity.

**Strengths:**

1. The proposed method is fully implemented through prompt engineering and in-context learning, without requiring any fine-tuning of the model, which significantly reduces computational cost and implementation complexity.
2. The disguise features are expressed in natural language form, providing interpretability and helping to reveal the intrinsic differences between AIGT and HWT.
3. The authors conduct a comprehensive evaluation using multiple quantitative metrics, thoroughly assessing both the effectiveness of the proposed attack and the quality of the generated texts.

**Weaknesses:**

1. The proposed approach shows a strong dependency on prompt design and the chosen proxy detector, which raises concerns about its generalizability to broader domains or unseen detection models.
2. The experimental setup is limited, using only 1,000 samples extracted from the RAID dataset and focusing solely on summarization tasks; further validation on diverse tasks and datasets is necessary to demonstrate robustness.
3. The external knowledge base entirely relies on the disguised texts generated during the feature extraction phase, resulting in a single-source dataset that may limit its quality and diversity, while also incurring a high maintenance cost.
4. The paper’s presentation is verbose, with key ideas not clearly emphasized; in particular, the mechanism by which the knowledge base improves text quality and diversity is not well explained.

**Questions:**

Please refer to my comments on weaknesses.

---

### Official Review · Reviewer_xQpQ · 2025-10-29

**Soundness:** 3
**Presentation:** 3
**Contribution:** 3
**Rating:** 6
**Confidence:** 4

**Summary:**

The paper proposes Self-Disguise Attack (SDA), a novel method for AI-generated text (AIGT) detection evasion designed to overcome the high fine-tuning costs and text degradation. SDA enables an LLM to actively disguise its outputs as more human-like through prompting rather than post-editing. It has two key components: an adversarial feature extractor that learns salient “disguise features” and a retrieval-based context examples optimizer that fetches relevant undetectable example texts from an external knowledge base. These components are combined into a single prompt containing the extracted disguise features and optimized in-context examples to guide the LLM in generating detection-resistant text. Extensive experiments show the effectiveness.

**Strengths:**

1. The paper is well written.
2. The idea is novel.

**Weaknesses:**

1. SDA’s effectiveness hinges on the proxy detector used to extract disguise features, which may limit its universality. It is tuned against a specific surrogate detector (e.g. a RoBERTa-based ChatGPT detector). It generalizes to other detectors better than baselines, but lack  explanation for this phenomenon.
2. SDA involves an iterative feature extraction process and the creation of an external knowledge base of disguised examples. While the experimental section provides some runtime measurements, I recommend that the authors include a more detailed analysis of the method’s computational complexity. Specifically, why it is more efficient than SICO?
3. The baselines are too weak. Can SDA evade zero-shot detectors like Fast-DetectGPT [R1], Binoculars [R2].

[R1]. Bao, G., Zhao, Y., Teng, Z., Yang, L., & Zhang, Y. (2023). Fast-detectgpt: Efficient zero-shot detection of machine-generated text via conditional probability curvature. arXiv preprint arXiv:2310.05130.

[R2]. Hans, A., Schwarzschild, A., Cherepanova, V., Kazemi, H., Saha, A., Goldblum, M., ... & Goldstein, T. (2024). Spotting llms with binoculars: Zero-shot detection of machine-generated text. arXiv preprint arXiv:2401.12070.

**Questions:**

See the weaknesses.

---

### Official Review · Reviewer_yTBL · 2025-10-31

**Soundness:** 3
**Presentation:** 3
**Contribution:** 2
**Rating:** 6
**Confidence:** 4

**Summary:**

Authors introduce Self-Disguise Attack (SDA), a novel approach that enables large language models (LLMs) to actively disguise their output, reducing the detection probability of AI Generated Text.  The method used to accomplish this is to ask the LLM generating text to mimic the patterns found in Human Generated text.

**Strengths:**

1. Using extracted features to guide generation is a good approach, that makes intuitive sense to evade detection
2. Using a large dataset to extract features makes this process automated and more reliable.
2. SDA shows better detection evasion results than existing AIGT methods and displays better generation quality

**Weaknesses:**

1. Distinction form prior work: Computation benefits are listed as one of the main benefits over existing methods,  but no statistics are given to support the claim
2. Lack of clarity: From fig 1, training data consists of detection evaded text, but from section 4.1 datasets is human generated text instead

**Questions:**

1. What are the computational benefits of SDA over previous methods? Two major benefits listed over previous methods are generation quality and generation cost. There is a table that displays generation quality improvement, but paper lacks results for generation resource consumption.
2. How is the data used to extract features gathered? Is it human generated text, or generated text that evades detection in prior checks? There appears to be an incongruity in Fig 1 and section 4.1. Adding clarity in these sections would enable better understanding and ensure reproducibility.

---

### Note · Authors · 2025-11-15

I have read and agree with the venue's withdrawal policy on behalf of myself and my co-authors.